# Minimax Lower Bounds for Estimating Distributions on Low-dimensional Spaces

**Saptarshi Chakraborty**          *saptarshic@berkeley.edu*
*Department of Statistics*
*University of California, Berkeley*

**Reviewed on OpenReview:** *https://openreview.net/forum?id=wIgRV336hC*

## Abstract

Recent statistical analyses of Generative Adversarial Networks (GAN) suggest that the error in estimating the target distribution in terms of the $\beta$-Hölder Integral Probability Metric (IPM) scales as $\mathcal{O}\left(n^{-\frac{\beta}{\overline{d}_{\mathbb{M}}+\delta}} \vee n^{-1/2} \log n\right)$. Here $\overline{d}_{\mathbb{M}}$ is the upper Minkowski dimension of the corresponding support $\mathbb{M}$ of the data distribution and $\delta$ is a positive constant. It is, however, unknown as to whether this rate is minimax optimal, i.e. whether there are estimators that achieve a better test-error rate. This paper demonstrates that the minimax rate for estimating unknown distributions in the $\beta$-Hölder IPM on $\mathbb{M}$ scales as $\Omega\left(n^{-\frac{\beta}{\underline{d}_{\mathbb{M}}-\delta}} \vee n^{-1/2}\right)$, where $\underline{d}_{\mathbb{M}}$ is the lower Minkowski dimension of $\mathbb{M}$. Thus if the low-dimensional structure $\mathbb{M}$ is regular in the Minkowski sense, i.e. $\overline{d}_{\mathbb{M}} = \underline{d}_{\mathbb{M}}$, GANs are roughly minimax optimal in estimating distributions on $\mathbb{M}$. Further, the paper shows that the minimax estimation rate in the $p$-Wasserstein metric scales as $\Omega\left(n^{-\frac{1}{\underline{d}_{\mathbb{M}}-\delta}} \vee n^{-1/(2p)}\right)$.

Nonparametric density estimation, aimed at approximating a probability distribution from a finite collection of identically and independently distributed (i.i.d.) samples, holds extensive application in the realms of statistics and machine learning. Nonparametric density estimation finds application in various fields such as mode estimation (Parzen, 1962), nonparametric classification (Rigollet, 2007; Chaudhuri et al., 2008), Monte Carlo computational methods (Doucet et al., 2001), and clustering (Chaudhuri & Dasgupta, 2010; Chakraborty et al., 2021; Rinaldo & Wasserman, 2010), among others. Typical techniques for nonparametric density estimation encompass the histogram method, kernel methods, k-nearest Neighbors (kNN) (Devroye & Wagner, 1977; Bhattacharya & Mack, 1987; Zhao & Lai, 2022), wavelet-based methods (Donoho et al., 1996) and more. Notably, the recent advancements in deep learning have led to the groundbreaking concept of Generative Adversarial Networks (GANs) (Goodfellow et al., 2014), which has revolutionised the field of nonparametric density estimation to obtain superhuman performance, especially for handling vision data.

The empirical successes of GANs have motivated researchers to study their theoretical guarantees. Biau et al. (2020) analyzed the asymptotic properties of vanilla GANs along with parametric rates. Biau et al. (2021) also analyzed the asymptotic properties of WGANs. Liang (2021) explored the min-max rates for WGANs for different non-parametric density classes and under a sampling scheme from a kernel density estimate of the data distribution; while Schreuder et al. (2021) studied the finite-sample rates under adversarial noise. Uppal et al. (2019) derived the convergence rates for Besov discriminator classes for WGANs. Luise et al. (2020) conducted a theoretical analysis of WGANs under an optimal transport-based paradigm. Recently, Asatryan et al. (2023) and Belomestny et al. (2021) improved upon the works of Biau et al. (2020) to understand the behaviour of GANs for Hölder class of density functions. Arora et al. (2017) showed that generalisation might not hold in standard metrics. However, they show that under a restricted "neural-net distance", the GAN is indeed guaranteed to generalize well. Recently, Arora et al. (2018) showed that GANs and their variants might not be well-equipped against mode collapse.

Although significant progress has been made in our theoretical understanding of GAN, some limitations of the existing results are yet to be addressed. For instance, the generalisation bounds frequently suffer from the curse of dimensionality. In practical applications, data distributions tend to have high dimensionality, making the convergence rates that have been proven exceedingly slow. However, high-dimensional data, such as images, texts, and natural languages, often possess latent low-dimensional structures that reduce the complexity of the problem. For example, it is hypothesised that natural images lie on a low-dimensional structure, in spite of its high-dimensional pixel-wise representation (Pope et al., 2020). Though in classical statistics there have been various approaches, especially using kernel tricks and Gaussian process regression that achieve a fast rate of convergence that depends only on their low intrinsic dimensionality (Bickel & Li, 2007; Kim et al., 2019), such results are largely unexplored in the context of GANs. Recently, Huang et al. (2022) expressed the generalisation rates for GAN when the data has low-dimensional support in the Minkowski sense and the latent space is one-dimensional; while Dahal et al. (2022) derived the convergence rates under the Wasserstein-1 distance in terms of the manifold dimension. It is important to note that the compact Riemannian manifold assumption of the support of the target distribution and the assumption of a bounded density of the target distribution on this manifold by Dahal et al. (2022) is a very strong assumption that might not hold in practice.

Despite these recent advances, it remains uncertain whether GAN estimates of the target distribution are optimal in the minimax sense for estimating distributions that are supported on a low-dimensional structure. The proposed analysis addresses this gap in the current literature by providing a comprehensive analysis of the minimax lower bound for estimating low-dimensional distributions and demonstrate that when $n$ i.i.d. samples are available from any target distribution on a set $\mathbb{M}$, the convergence rate for any estimator (in the $\beta$-Hölder IPM) is at least $\Omega\left(n^{-\frac{\beta}{\underline{d}_{\mathbb{M}}-\delta}} \vee n^{-1/2}\right)$, where $\delta$ is a positive constant in the range of $(0, \underline{d}_{\mathbb{M}})$, and $\underline{d}_{\mathbb{M}}$ is the lower Minkowski dimension of the set $\mathbb{M}$. Thus, when the set $\mathbb{M}$ is Minkowski regular, GANs almost match this rate, when the networks are properly chosen. In particular, Huang et al. (2022) showed that when the generator and discriminators are realized by a feed-forward neural network with ReLU activation, with respective depths $L_{\mathrm{gen}}$, $L_{\mathrm{dis}}$ and widths $W_{\mathrm{gen}}$, $W_{\mathrm{dis}}$, then one can choose $W_{\mathrm{gen}}^2 L_{\mathrm{gen}} \precsim n$ and $W_{\mathrm{dis}} L_{\mathrm{dis}} \precsim n^{D/(2\bar{d}_{\mathbb{M}}+2\delta)} \log^2 n$ ($D$ denotes the dimension of the data space) to ensure that the error rate for the GAN estimate of the distribution scales as $\mathcal{O}(n^{-\beta/(\bar{d}_{\mathbb{M}}+\delta)} \vee n^{-1/2} \log n)$. In this case, the discriminator network has to satisfy a regularity condition in terms of its maximum Lipschitz constant. I refer the reader to Theorem 19 of Huang et al. (2022) for more details. Additionally, this work demonstrates that the minimax estimation rate in the $p$-Wasserstein metric decreases in proportion to $\Omega\left(n^{-\frac{1}{\underline{d}_{\mathbb{M}}-\delta}} \vee n^{-1/(2p)}\right)$.

# 1 Background

## 1.1 Related Work

Recent research has explored the minimax rates under the Wasserstein distances under various settings. Singh & Póczos (2018) demonstrated the minimax convergence rates assuming the distribution is compactly supported. In a related context, Liang (2021) and Uppal et al. (2019) established minimax convergence rates for the Wasserstein-1 distance under a smoothness assumption on the corresponding density. It has been demonstrated that estimating under the Integral Probability Metric (IPM) with smooth functions can lead to enhanced rates of convergence for empirical measures Kloeckner (2020). Niles-Weed & Berthet (2022) established the minimax convergence rates for Besov densities for the Wasserstein-$p$ metric. For smooth densities, the derived minimax rates can be improved (McDonald, 2017; Liang, 2021) in the sense that estimating a smooth density is easier than estimating a non-smooth one. However, all the aforementioned findings primarily consider the minimax rates when the corresponding distribution varies across all probability measures on a compact set, resulting in rates of $\mathcal{O}(n^{-1/D})$ or similar for estimating distributions of $\mathbb{R}^D$. Recently, Tang & Yang (2023) derived the minimax rates when the data is supported on a smooth sub-manifold with a positively bounded reach, and has a smooth density w.r.t. the volume measure on this manifold. In contrast to the work of Tang & Yang (2023), this analysis does not impose a smooth manifold structure on the data support, allowing for highly non-smooth and irregular geometries. Moreover, this work does not assume the existence of a density for the distribution on this potentially irregular low-dimensional

Table 1: A comparison of different upper and lower bounds for distribution estimation on the $D$-dimensional unit hypercube ($D \geq 3$). The upper bounds do not show possible poly-log factors in $n$.

| Result | Assumption on Distribution Support | Assumption on Density | Result Type | Metric | Bound |
|---|---|---|---|---|---|
| Liang (2021) | None | $\alpha$-Hölder | Upper and Lower Bound | $\beta$-Hölder IPM | $n^{-\frac{\alpha+\beta}{2\alpha+D}} \vee n^{-1/2}$ |
| Niles-Weed & Berthet (2022) | None | $s$-Besov | Upper and Lower Bound | $p$-Wasserstein | $n^{-\frac{1+s/p}{D+s}} \vee n^{-1/2p}$ |
| Huang et al. (2022) | Minkowski dimension $d$ | None | Upper Bound | $\beta$-Hölder IPM | $n^{-\beta/(d+\delta)} \vee n^{-1/2}$ |
| Tang & Yang (2023) | $d$-dimensional $\gamma$-smooth ($\gamma \geq 2$) sub-manifold with a lower-bounded reach | $\alpha$-Hölder and lower bounded | Upper and Lower Bound | $\beta$-Hölder IPM | $n^{-\frac{\beta\gamma}{2\alpha+d}} \vee n^{-\frac{\alpha+\beta}{2\alpha+d}} \vee n^{-1/2}$ |
| This work | Minkowski dimension $d$ | None | Lower Bound | $\beta$-Hölder IPM | $n^{-\beta/(d-\delta)} \vee n^{-1/2}$ |
| | | | | $p$-Wasserstein | $n^{-1/(d-\delta)} \vee n^{-1/(2p)}$ |

support, enabling us to accommodate singular distributions on this structure within the proposed framework. While both this work and the one by Tang & Yang (2023) leverage the principles of minimax error bounds via Fano's method, their reliance on smooth manifolds and the existence of smooth densities simplifies the corresponding testing problem for rate derivation. In contrast, the proposed approach circumvents these assumptions by constructing point measures on the low-dimensional structure, enabling a more general analysis under minimal assumptions. Further, while Tang & Yang (2023) derive the minimax error bounds for $\beta$-Hölder IPMs, the proposed analyses also cover Wasserstein $p$-distances. Table 1 provides a comparison of error rates from several prominent works in the literature, contextualizing the findings of this paper within the broader scope of existing research.

In addition to the mathematical results (Huang et al., 2022) that the error rates for GANs depend on the intrinsic data dimension, empirical results also indicate such phenomena occur in practice. For example, the recent works by Chakraborty & Bartlett (2024a) show that the test error rates in the Wasserstein-1 distance for Wasserstein Autoencoders (Tolstikhin et al., 2018) depend only on the Minkowski dimension of the data support. Similar results have also been established both theoretically and empirically for deep regression (Nakada & Imaizumi, 2020) and federated learning models (Chakraborty & Bartlett, 2024b).

## 1.2 Preliminaries and Notations

Before going into the details of the theoretical results, let us introduce some notation and recall some preliminary concepts.

We will use the notation $x \vee y := \max\{x, y\}$. $B_\varrho(x, r)$ denotes the open ball of radius $r$ around $x$, with respect to (w.r.t.) the metric $\varrho$. For any measure $\gamma$, the support of $\gamma$ is defined as, $\text{supp}(\gamma) = \{x : \gamma(B_\varrho(x, r)) > 0, \text{ for all } r > 0\}$. For any function $f : S \to \mathbb{R}$, and any measure $\gamma$ on $S$, let $\|f\|_{\mathbb{L}_p(\gamma)} := \left(\int_S |f(x)|^p d\gamma(x)\right)^{1/p}$, if $0 < p < \infty$. Also let, $\|f\|_{\mathbb{L}_\infty(\gamma)} := \text{ess sup}_{x \in \text{supp}(\gamma)} |f(x)|$. We say $A_n \precsim B_n$ (also written as $A_n = \mathcal{O}(B_n) \iff B_n = \Omega(A_n)$) if there exists $C > 0$, independent of $n$, such that $A_n \leq C B_n$. We say $A_n \asymp B_n$, if $A_n \precsim B_n$ and $B_n \precsim A_n$. For any $k \in \mathbb{N}$, we let $[k] = \{1, \ldots, k\}$. $\mathbb{1}(\cdot)$ denotes the indicator function. For any measure $\mu$, $\mu^{\otimes n}$ denotes the $n$-product measure of $\mu$. Let us also recall some useful definitions as follows.

**Definition 1** (Covering and Packing Numbers). For a metric space $(S, \varrho)$, the $\epsilon$-covering number w.r.t. $\varrho$ is defined as: $\mathcal{N}(\epsilon; S, \varrho) = \inf\{n \in \mathbb{N} : \exists x_1, \ldots x_n \text{ such that } \cup_{i=1}^n B_\varrho(x_i, \epsilon) \supseteq S\}$. Similarly, the $\epsilon$-packing number is defined as: $\mathcal{M}(\epsilon; S, \varrho) = \sup\{m \in \mathbb{N} : \exists x_1, \ldots x_m \in S \text{ such that } \varrho(x_i, x_j) \geq \epsilon, \text{ for all } i \neq j\}$.

**Definition 2** (Hölder functions). Let $f : S \to \mathbb{R}$ be a function, where $S \subseteq \mathbb{R}^D$. For a multi-index $\boldsymbol{s} = (s_1, \ldots, s_D)$, let, $\partial^{\boldsymbol{s}} f = \frac{\partial^{|\boldsymbol{s}|} f}{\partial x_1^{s_1} \ldots \partial x_D^{s_D}}$, denote the weak partial derivative of $f$, where, $|\boldsymbol{s}| = \sum_{\ell=1}^{D} s_\ell$. A function $f : S \to \mathbb{R}$ is said to be $\beta$-Hölder (for $\beta > 0$) if

$$\|f\|_{\mathbb{H}^\beta} := \sum_{\boldsymbol{s}: 0 \le |\boldsymbol{s}| \le \lfloor \beta \rfloor} \|\partial^{\boldsymbol{s}} f\|_\infty + \sum_{\boldsymbol{s}: |\boldsymbol{s}| = \lfloor \beta \rfloor} \sup_{x \ne y} \frac{\|\partial^{\boldsymbol{s}} f(x) - \partial^{\boldsymbol{s}} f(y)\|}{\|x - y\|^{\beta - \lfloor \beta \rfloor}} < \infty.$$

If $f : \mathbb{R}^D \to \mathbb{R}^{D'}$, then define $\|f\|_{\mathbb{H}^\beta} = \sum_{j=1}^{D'} \|f_j\|_{\mathbb{H}^\beta}$. For notational simplicity, let, $\mathbb{H}^\beta(S_1, S_2, C) = \{f : S_1 \to S_2 : \|f\|_{\mathbb{H}^\beta} \le C\}$. Here, both $S_1$ and $S_2$ are subsets of real vector spaces. If $S_1 = [0,1]^D$, $S_2 = \mathbb{R}$ and $C = 1$, we write $\mathbb{H}^\beta$ in stead of $\mathbb{H}^\beta(S_1, S_2, C)$.

Next, let us recall the definitions of Total Variation and Wasserstein-$p$ distances as well as Integral Probability Metrics (IPMs).

**Definition 3** (Total Variation Distance). Let $\Omega$ be a Polish space and suppose that $\mu$ and $\nu$ are two probability measures defined on $\Omega$. Then, the total variation distance between $\mu$ and $\nu$ is defined as,

$$\mathrm{TV}(\mu, \nu) = \sup_{B \in \mathscr{B}(\Omega)} |\mu(B) - \nu(B)| = \inf_{\gamma \in \Gamma(\mu,\nu)} \mathbb{P}_{(X,Y) \sim \gamma}(X \ne Y). \tag{1}$$

Here, $\mathscr{B}(\Omega)$ denotes the Borel $\sigma$-algebra on $\Omega$ and $\Gamma(\mu, \nu)$ denotes the set of all measure couples between $\mu$ and $\nu$. The reader is referred to Proposition 4.7 of Levin & Peres (2017) for a proof of the second equality in (1).

**Definition 4** (Wasserstein $p$-distance). Let $(\Omega, \mathrm{dist})$ be a Polish space and let $\mu$ and $\nu$ be two probability measures on the same with finite $p$-moments. Then the $p$-Wasserstein distance between $\mu$ and $\nu$ is defined as:

$$\mathbb{W}_p(\mu, \nu) = \left( \inf_{\gamma \in \Gamma(\mu,\nu)} \mathbb{E}_{(X,Y) \sim \gamma} \left( \mathrm{dist}(X, Y) \right)^p \right)^{1/p}.$$

In what follows, $\mathrm{dist}(\cdot, \cdot)$ is taken to be the $\ell_2$-norm on $\mathbb{R}^D$.

**Definition 5** (Integral Probability Metric). For a function class $\mathcal{F}$, the $\mathcal{F}$-Integral Probability Metric (IPM) between two probabiloty measures $\mu$ and $\nu$ is defined as,

$$\|\mu - \nu\|_{\mathcal{F}} = \sup_{f \in \mathcal{F}} \left| \int f d\mu - \int f d\nu \right|.$$

### 1.3 Minkowski Dimension

Often, real data is hypothesized to lie on a lower-dimensional structure within the high-dimensional representative feature space. To characterize this low-dimensionality of the data, researchers have defined various notions of the effective dimension of the underlying measure from which the data is assumed to be generated. Among these approaches, the most popular ones use some sort of rate of increase of the covering number, in the log-scale, of most of the support of this data distribution. Let $(S, \varrho)$ be a compact Polish space and let $\mu$ be a probability measure defined on it. Throughout the remainder of the paper, $\varrho$ is taken to be the $\ell_\infty$-norm. The low-dimensional nature of the data distribution is characterised by the Minkowski dimension of the support of $\mu$. Let us recall the definition of Minkowski dimensions (Falconer, 2004),

**Definition 6** (Minkowski dimension). *For a bounded metric space $(S, \varrho)$, the upper Minkwoski dimension of $S$ is defined as*

$$\overline{d}_S = \limsup_{\epsilon \downarrow 0} \frac{\log \mathcal{N}(\epsilon; S, \varrho)}{\log(1/\epsilon)}.$$

*Similarly, the lower Minkowski dimension of $S$ is given by,*

$$\underline{d}_S = \liminf_{\epsilon \downarrow 0} \frac{\log \mathcal{N}(\epsilon; S, \varrho)}{\log(1/\epsilon)}.$$

*If $\underline{d}_S = \overline{d}_S$, we say that $S$ is Minkowski regular and a has Minkowski dimension of $d_S = \lim_{\epsilon \downarrow 0} \frac{\log \mathcal{N}(\epsilon; S, \varrho)}{\log(1/\epsilon)}$.*

The Minkowski dimension essentially measures how the covering number of $S$ is affected by the radius of the covering balls. Since this notion of dimensionality depends only on the covering numbers and does not assume the existence of a smooth correspondence to a smaller dimensional Euclidean space, this notion not only incorporates smooth manifolds but also covers highly non-smooth sets such as fractals. In the literature, Kolmogorov & Tikhomirov (1961) provided a comprehensive study on the dependence of the covering number of different function classes on the underlying Minkowski dimension of the support. Nakada & Imaizumi (2020) showed how deep learners can incorporate this low-dimensionality of the data that is also reflected in their convergence rates. Recently, Huang et al. (2022) showed that WGANs can also adapt to this low-dimensionality of the data. In particular they showed that when the data is independent and identically distributed from a distribution $\mu$, for the GAN estimate for the density (denoted as $\hat{\mu}^{\mathrm{GAN}}$), $\|\mu - \hat{\mu}^{\mathrm{GAN}}\|_{\mathbb{H}^\beta}$ decays at a rate of $\mathcal{O}\left(n^{-\frac{\beta}{\bar{d}_{\mathbb{M}}+\delta}} \vee n^{-1/2}\log n\right)$, where $\mathbb{M}$ is the support of $\mu$ and $\delta > 0$ is a pre-fixed constant. In the following section, we attempt to understand whether this rate is optimal or not.

## 2 Theoretical Analysis

Suppose that $\mathbb{M} \subseteq [0,1]^D$ and let $\Pi_{\mathbb{M}}$ denote the set of all probability distributions on $\mathbb{M}$. Let us assume that one has access to $n$ samples, $X_1, \ldots, X_n$, generated independently from $\mu \in \Pi_{\mathbb{M}}$. The goal is to understand how well any estimate of $\mu$, based on the data, performs. This performance is charecterised in terms of the $\beta$-Hölder IPM or the $p$-Wasserstein distance, i.e. for an estimate $\hat{\mu}$, its performance is measured as $\|\mu - \hat{\mu}\|_{\mathbb{H}^\beta}$ or $\mathbb{W}_p(\hat{\mu}, \mu)$. To characterise this notion of best-performing estimator, researchers use the concept of minimax risk i.e. the risk of the best-performing estimator that achieves the minimum risk with respect to all members in $\Pi_{\mathbb{M}}$. Formally, the minimax risk for the problem is given by,

$$\mathfrak{M}_n = \inf_{\hat{\mu}} \sup_{\mu \in \Pi_{\mathbb{M}}} \mathbb{E}_\mu \|\hat{\mu} - \mu\|_{\mathbb{H}^\beta} \quad \text{or} \quad \mathfrak{M}_n = \inf_{\hat{\mu}} \sup_{\mu \in \Pi_{\mathbb{M}}} \mathbb{E}_\mu \mathbb{W}_p(\hat{\mu}, \mu),$$

where the infimum is taken over all measurable estimates of $\mu$, i.e. on $\{\hat{\mu} : (X_1, \ldots, X_n) \to \Pi_{[0,1]^D} : \hat{\mu} \text{ is measurable}\}$. Here, $\mathbb{E}_\mu$ is used to denote that the expectation is taken with respect to the joint distribution of $X_1, \ldots, X_n$, which are independently and identically distributed as $\mu$. Theorem 7 states the main lower bound of this paper, which lower bounds $\mathfrak{M}_n$ in terms of the lower-Minkowski dimension of $\mathbb{M}$ and the number of samples $n$, when $n$ is large. The proof of this result is given in Section 3.

**Theorem 7** (Main Result). *Suppose that $X_1, \ldots, X_n$ are i.i.d. $\mu$ and let $\delta \in (0, \underline{d}_{\mathbb{M}})$. Then there exists an $n_0 \in \mathbb{N}$ such that if $n \geq n_0$,*

$$\inf_{\hat{\mu}} \sup_{\mu \in \Pi_{\mathbb{M}}} \mathbb{E}_\mu \|\hat{\mu} - \mu\|_{\mathbb{H}^\beta} \gtrsim n^{-\frac{\beta}{\underline{d}_{\mathbb{M}}-\delta}} \vee n^{-1/2}, \tag{2}$$

*where the infimum is taken over all measurable estimates of $\mu$, based on the data, $X_1, \ldots, X_n$. Furthermore,*

$$\inf_{\hat{\mu}} \sup_{\mu \in \Pi_{\mathbb{M}}} \mathbb{E}_\mu \mathbb{W}_p(\hat{\mu}, \mu) \gtrsim n^{-\frac{1}{\underline{d}_{\mathbb{M}}-\delta}} \vee n^{-\frac{1}{2p}}. \tag{3}$$

If $\mathbb{M}$ is Minkowski regular, from Theorem 7, note that for any $\delta \in (0, d_{\mathbb{M}})$,

$$\inf_{\hat{\mu}} \sup_{\mu \in \Pi_{\mathbb{M}}} \mathbb{E}_\mu \|\hat{\mu} - \mu\|_{\mathbb{H}^\beta} \gtrsim n^{-\frac{\beta}{d_{\mathbb{M}}-\delta}} \vee n^{-1/2}.$$

From the results derived by Huang et al. (2022), GANs can achieve a rate of convergence of $\mathcal{O}\left(n^{-\frac{\beta}{d_{\mathbb{M}}+\delta}} \vee n^{-1/2}\log n\right)$, implying that GANs are almost optimal in learning distributions when the data is low-dimensional in the Minkowski sense, barring poly-log factors in the sample size.

It is important to note that the lower bound in (3) closely resembles the ones derived by Niles-Weed & Berthet (2022) for distributions with Besov densities. Furthermore, from Theorem 1 of Weed & Bach (2019), note that the empirical distribution $\hat{\mu}_n$ scales as $\mathbb{E}W_p(\hat{\mu}_n, \mu) \precsim n^{-1/(d_p^*(\mu)+\delta)}$, where $d_p^*(\mu)$ denotes the $p$-upper Wasserstein dimension of $\mu$. Since $\mu$ is supported on $\mathbb{M}$, by Proposition 2 of Weed & Bach (2019), $d_p^*(\mu) \leq \bar{d}_{\mathbb{M}}$,

when $\underline{d}_{\mathbb{M}} \geq 2p$. Thus, $\mathbb{E} W_p(\hat{\mu}_n, \mu) \precsim n^{-1/(\bar{d}_{\mathbb{M}}+\delta)}$. Hence, when $\underline{d}_{\mathbb{M}} > 2p$, we can choose $\delta > 0$, such that, $\inf_{\hat{\mu}} \sup_{\mu \in \Pi_{\mathbb{M}}} \mathbb{E}_{\mu} \mathbb{W}_p(\hat{\mu}, \mu) \succsim n^{-1/(\underline{d}_{\mathbb{M}}-\delta)}$. Hence, in this case, the empirical distribution almost achieves this minimax optimal rate when $\underline{d}_{\mathbb{M}} > 2p$ and $\mathbb{M}$ is Minkowski regular.

**Remark 1** ($\delta$-term in Theorem 7). We observe that the $\delta$-term is an artefact of the definition of the Minkowski dimension. If $\mathcal{N}(\epsilon, \mathbb{M}, \ell_\infty) \succsim \epsilon^{-\underline{d}}$, for some $\underline{d} \in (0, D]$, i.e. when the limit in the computation of the lower Minkowski dimension can be achieved exactly, following the proof of Theorem 7 (see Section 3), note that under the same assumptions $\mathfrak{M}_n \succsim n^{-\beta/\underline{d}} \vee n^{-1/2}$, for $n$ large. The $\delta$-term in the lower bound is only an artefact of the definition of the lower Minkowski dimension and can be removed by assuming the lower bound for the covering number. Further, if one assumes that $\mathcal{N}(\epsilon, \mathbb{M}, \ell_\infty) \precsim \epsilon^{-\bar{d}}$, then the analyses by Huang et al. (2022) shows that the error rate for GANs for estimating a distribution $\mu$, supported on $\mathbb{M}$ scales as $(n^{-\beta/\bar{d}} \vee n^{-1/2}) \log n$ (see Theorem 19 of Huang et al. (2022)). Thus, when $\mathcal{N}(\epsilon, \mathbb{M}, \ell_\infty) \asymp \epsilon^{-d}$, the error rates for GANs match the minimax rate except for an excess log-factor in the number of samples.

**Inference for distributions supported on a Manifold**   When the support is regular, one can say that the minimax rate for estimating distributions decays at a rate whose exponent is inversely proportional to its regularity dimension. We recall that a set $\mathbb{M}$ is $d$-regular w.r.t. the $d$-dimensional Hausdorff measure $\mathscr{H}^d$ if

$$\mathscr{H}^d(B_\varrho(x, r)) \asymp r^d,$$

for all $x \in \mathbb{M}$ (see Definition 6 of Weed & Bach (2019)). Recall that the $d$-Hausdorff measure of a set $S$ is defined as,

$$\mathscr{H}^d(S) := \liminf_{\epsilon \downarrow 0} \left\{ \sum_{k=1}^\infty r_k^d : S \subseteq \sum_{k=1}^\infty B_\varrho(x_k, r_k), r_k \leq \epsilon, \forall k \right\}.$$

It is known (Mattila, 1999) that if $\mathbb{M}$ is $d$-regular, then $d_{\mathbb{M}} = d$. Thus, when $\mathbb{M}$ is $d$-regular, the minimax rate roughly scales at $\Omega(n^{-\beta/d})$. Since, compact $d$-dimensional differentiable manifolds are $d$-regular (Weed & Bach, 2019, Proposition 9), this implies that for when $\mathbb{M}$ is a compact differentiable $d$-dimensional manifold, the error rates scale as $\Omega(n^{-\beta/d})$. This result underscores that GANs are nearly minimax optimal, given that the corresponding upper bounds derived by Dahal et al. (2022) match this minimax rate, albeit under some additional assumptions regarding the smoothness of the manifold. A similar result holds when $\mathbb{M}$ is a nonempty, compact convex set spanned by an affine space of dimension $d$; the relative boundary of a nonempty, compact convex set of dimension $d + 1$; or a self-similar set with similarity dimension $d$ as all these sets are $d$-regular by (Weed & Bach, 2019, Proposition 9).

## 3   Proof of the Main Result (Theorem 7)

As a first step for deriving a minimax bound, let first show that the Hölder IPM can be lower bounded by the total variation distance and the minimum separation of the support of the distributions. For any finite set, we use the notation, $\text{sep}(\Xi) = \inf_{\xi, \xi' \in \Xi : \xi \neq \xi'} \|\xi - \xi'\|_\infty$.

**Lemma 8.** *Let $\Xi$ be a finite subset of $\mathbb{R}^d$ and let, $P, Q \in \Pi_\Xi$. Then, there exists a constant $\pi_1$ (that might depend on $\beta$) such that, $\|P - Q\|_{\mathbb{H}^\beta(\mathbb{R}^D, \mathbb{R}, 1)} \geq \pi_1 (sep(\Xi))^\beta \|P - Q\|_{TV}$.*

*Proof.* Let $b(x) = \exp\left(\frac{1}{x^2-1}\right) \mathbb{1}\{|x| \leq 1\}$ be the standard bump function on $\mathbb{R}$. For any $x \in \mathbb{R}^D$ and $\delta \in (0, 1]$, let, $h_\delta(x) = a\delta^\beta \prod_{j=1}^d b(x_j/\delta)$. Here $a$ is such that $ab(x) \in \mathbb{H}^\beta(\mathbb{R}, \mathbb{R}, C)$. It is easy to observe that $h_\delta \in \mathbb{H}^\beta(\mathbb{R}^D, \mathbb{R}, 1)$. Let $P$ and $Q$ be two distributions on $\Xi = \{\xi_1, \ldots, \xi_k\}$. Let $\delta = \frac{1}{3} \min_{i \neq j} \|\xi_i - \xi_j\|_\infty$. Define $h^\star(x) = \sum_{i=1}^k \alpha_i h_\delta(x - \xi_i)$, with $\alpha_i \in \{-1, +1\}$, to be chosen later. Since the individual terms in $h^\star$ are members of $\mathbb{H}^\beta(\mathbb{R}^D, \mathbb{R}, 1)$ and have disjoint supports, $h^\star \in \mathbb{H}^\beta(\mathbb{R}^D, \mathbb{R}, 1)$. Taking $\alpha_i = 2\mathbb{1}(P(\xi) \geq Q(\xi)) - 1$, it is easy to see that,

$$\|P - Q\|_{\mathbb{H}^\beta(\mathbb{R}^D, \mathbb{R}, 1)} \geq \int h^* dP - \int h^* dQ = \sum_{i=1}^k a\delta^\beta \alpha_i (P(\xi_i) - Q(\xi_i)) = a\delta^\beta \sum_{i=1}^k |P(\xi_i) - Q(\xi_i)|$$

$$= 2a\delta^\beta \|P - Q\|_{\text{TV}}$$

$$= \frac{2a}{3^\beta}(\text{sep}(\Xi))^\beta \|P - Q\|_{\text{TV}}.$$

Taking $\pi_1 = \frac{2a}{3^\beta}$ gives us the desired result. $\qquad\square$

Similar to Lemma 8, on a discrete space, the $p$-Wasserstein metric is lower bounded by the total variation distance.

**Lemma 9.** *Let $\Xi$ be a finite subset of $\mathbb{R}^d$ and let, $P, Q \in \Pi_\Xi$. Then, $\mathbb{W}_p(P, Q) \geq sep(\Xi)\|P - Q\|_{TV}^{1/p}$.*

*Proof.* Let $X \sim P$ and $Y \sim Q$. Note that $\|X - Y\|_2 \geq \mathbb{1}\{X \neq Y\}\,\text{sep}(\Xi)$. Thus,

$$(\mathbb{E}\|X - Y\|_2^p)^{1/p} \geq (\mathbb{P}(X \neq Y))^{1/p}\,\text{sep}(\Xi).$$

Taking infimum w.r.t. all measure couples between $P$ and $Q$ gives us the desired result. $\qquad\square$

With the above two lemmas at our disposal, we are now ready to prove the main result of this paper. Recall that if $P \ll Q$, the KL-divergence between $P$ and $Q$ is given by, $\text{KL}(P\|Q) = \int \log(dP/dQ)dP$. Similarly, the $\chi^2$-divergence is given by, $\chi^2(P\|Q) = \int (dP/dQ)^2 - 1$.

### 3.1 Proof of Theorem 7

With Lemmata 8 and 9, we are now ready to prove Theorem 7. We use Fano's method to obtain the minimax lower bound. I refer the reader to Chapter 15 of Wainwright (2019) for a detailed exposition. Let, $s = \underline{d}_\mathbb{M} - \delta$. Thus, one can find $\epsilon_0 \in (0, 1)$, such that if $\epsilon \in (0, \epsilon_0]$, $\mathcal{N}(\epsilon, \mathbb{M}, \ell_\infty) \geq \epsilon^{-s} \implies \mathcal{M}(\epsilon, \mathbb{M}, \ell_\infty) \geq \epsilon^{-s}$. Take $n \geq n_0 = (128(\epsilon_0)^{-s}) \vee 8192$. Suppose $\epsilon = (n/128)^{-1/s}$. Let $\Theta = \{\theta_1, \ldots, \theta_k\}$ be a $\epsilon$-separated set in $\mathbb{M}$. For the above choices of $n$ and $\epsilon$, observe that one can take, $k = \epsilon^{-s} = n/128 \geq 64$ and $n \geq 64k$.

Let $\phi_j(x) = \mathbb{1}\{x = \theta_j\} - \mathbb{1}\{x = \theta_{\lfloor k/2 \rfloor + j}\}$, for all $j = 1, \ldots, \lfloor k/2 \rfloor$. Let, $\boldsymbol{\omega} \in \{0, 1\}^{\lfloor k/2 \rfloor}$. Define the probability mass function on $\Theta$,

$$P_{\boldsymbol{\omega}}(x) = \frac{1}{k} + \frac{\delta_k}{k} \sum_{j=1}^{\lfloor k/2 \rfloor} \omega_j \phi_j(x),$$

with $\delta_k \in (0, 1/2]$. By construction, $P_{\boldsymbol{\omega}} \in \Pi_\mathbb{M}$.

Furthermore,

$$\|P_{\boldsymbol{\omega}} - P_{\boldsymbol{\omega}'}\|_{\text{TV}} = \frac{\delta_k}{k}\|\boldsymbol{\omega} - \boldsymbol{\omega}'\|_1.$$

By the Varshamov-Gilbert bound (Tsybakov, 2009, Lemma 2.9), let $\Omega \subseteq \{0, 1\}^{\lfloor k/2 \rfloor}$ be such that $|\Omega| \geq 2^{\frac{1}{8}\lfloor k/2 \rfloor}$ and $\|\boldsymbol{\omega} - \boldsymbol{\omega}'\|_1 \geq \frac{1}{8}\lfloor k/2 \rfloor$, for all $\boldsymbol{\omega} \neq \boldsymbol{\omega}'$ both in $\Omega$. Thus for any $\boldsymbol{\omega} \neq \boldsymbol{\omega}'$, both in $\Omega$,

$$\|P_{\boldsymbol{\omega}} - P_{\boldsymbol{\omega}'}\|_{\text{TV}} \geq \frac{\delta_k \lfloor k/2 \rfloor}{8k}. \tag{4}$$

Hence, by Lemma 8, $\|P_{\boldsymbol{\omega}} - P_{\boldsymbol{\omega}'}\|_{\mathbb{H}^\beta(\mathbb{R}^D, \mathbb{R}, 1)} \geq \pi_1 \epsilon^\beta \frac{\delta_k \lfloor k/2 \rfloor}{k}$. Similarly, by Lemma 9, it is easy to note that $\mathbb{W}_p(P_{\boldsymbol{\omega}}, P_{\boldsymbol{\omega}'}) \geq \epsilon \left(\frac{\delta_k \lfloor k/2 \rfloor}{k}\right)^{1/p}$. Furthermore, observe that

$$
\begin{aligned}
\text{KL}(P_{\boldsymbol{\omega}}^{\otimes n}\|P_{\boldsymbol{\omega}'}^{\otimes n}) = n\,\text{KL}(P_{\boldsymbol{\omega}}\|P_{\boldsymbol{\omega}'}) \leq n\,\chi^2(P_{\boldsymbol{\omega}}\|P_{\boldsymbol{\omega}'}) = n\sum_{i=1}^{k} \frac{(P_{\boldsymbol{\omega}}(\xi_i) - P_{\boldsymbol{\omega}'(\xi_i)})^2}{P_{\boldsymbol{\omega}}(\xi_i)} &\leq 2nk\sum_{i=1}^{k}(P_{\boldsymbol{\omega}}(\xi_i) - P_{\boldsymbol{\omega}'}(\xi_i))^2 \\
&\leq 2nk\lfloor k/2 \rfloor (2\delta_k/k)^2 \\
&= 8\frac{n\lfloor k/2 \rfloor \delta_k^2}{k}.
\end{aligned}
$$

Thus,

$$\frac{1}{|\Omega|^2} \sum_{\boldsymbol{\omega},\boldsymbol{\omega}' \in \Omega} \mathrm{KL}(P_{\boldsymbol{\omega}}^{\otimes n} \| P_{\boldsymbol{\omega}'}^{\otimes n}) \leq 8 \frac{n \lfloor k/2 \rfloor \delta_k^2}{k}.$$

Let $\mathcal{P} = \{P_{\boldsymbol{\omega}} : \boldsymbol{\omega} \in \Omega\}$. Let $J \sim \mathrm{Unif}(\Omega)$ and $Z|J = \boldsymbol{\omega} \sim P_\omega$. By the convexity of KL divergence (see equation 15.34 of Wainwright (2019)), it is known that,

$$I(Z;J) \leq \frac{1}{|\Omega|^2} \sum_{\boldsymbol{\omega},\boldsymbol{\omega}' \in \Omega} \mathrm{KL}(P_{\boldsymbol{\omega}}^{\otimes n} \| P_{\boldsymbol{\omega}'}^{\otimes n}) \leq 8 \frac{n \lfloor k/2 \rfloor \delta_k^2}{k}.$$

Thus,

$$\frac{I(Z;J) + \log 2}{\log |\Omega|} \leq 8 \frac{I(Z;J) + \log 2}{\lfloor k/2 \rfloor \log 2} \leq 64 \frac{n \delta_k^2}{k \log 2} + \frac{8}{\lfloor k/2 \rfloor} \leq 64 \frac{n \delta_k^2}{k \log 2} + \frac{1}{4}. \tag{5}$$

The last inequality follows since $k \geq 64$. Take $\delta_k = \frac{1}{16}\sqrt{\frac{k \log 2}{n}}$. Clearly, $\epsilon \leq 1/2$ as $n \geq 64k$. This choice of $\epsilon$ makes,

$$\frac{I(Z;J) + \log 2}{\log |\Omega|} \leq \frac{1}{2}.$$

Thus, by Theorem 15.12 of Wainwright (2019) (taking $\Phi(x) = x$ and $\rho(P,Q) = \|P - Q\|_{\mathbb{H}^\beta}$),

$$\inf_{\hat{\mu}} \sup_{\mu \in \Pi_{\mathbb{M}}} \mathbb{E}_\mu \|\hat{\mu} - \mu\|_{\mathbb{H}^\beta} \geq \pi_1 \epsilon^\beta \frac{\delta_k \lfloor k/2 \rfloor}{k} = \pi_1 \epsilon^\beta \frac{\lfloor k/2 \rfloor}{k} \frac{1}{16} \sqrt{\frac{k \log 2}{n}} = \pi_1 \epsilon^\beta \frac{\lfloor k/2 \rfloor}{k} \frac{1}{16} \sqrt{\frac{\log 2}{128}}$$

$$\geq \pi_2 \epsilon^\beta \asymp n^{-\beta/s}. \tag{6}$$

$$\text{Similarly,} \quad \inf_{\hat{\mu}} \sup_{\mu \in \Pi_{\mathbb{M}}} \mathbb{E}_\mu \mathbb{W}_p(\hat{\mu}, \mu) \geq \epsilon \left( \frac{\delta_k \lfloor k/2 \rfloor}{k} \right)^{1/p} \gtrsim n^{-1/s}. \tag{7}$$

To show that $\inf_{\hat{\mu}} \sup_{\mu \in \Pi_{\mathbb{M}}} \|\hat{\mu} - \mu\|_{\mathbb{H}^\beta} \gtrsim n^{-1/2}$, one can use Le Cam's method (Wainwright, 2019, Chapter 15.2). Let $\theta_0, \theta_1 \in \mathbb{M}$ be such that $\|\theta_0 - \theta_1\|_\infty \geq \mathrm{diam}(\mathbb{M})/2$. Let $P_0(\theta_0) = P_0(\theta_1) = 1/2$ and $P_1(\theta_0) = 1 - P_1(\theta_1) = 1/2 - \delta$ with $\delta \in (0, 1/4)$. Clearly, $\mathrm{TV}(P_0, P_1) = \delta$. Thus, by Lemma 8, observe that

$$\|P_1 - P_0\|_{\mathbb{H}^\beta} \gtrsim (\mathrm{diam}(\mathbb{M})/2)^\beta \delta \gtrsim \delta.$$

Similarly, $\mathbb{W}_p(P_1, P_0) \geq (\mathrm{diam}(\mathbb{M})/2)\delta^{1/p} \gtrsim \delta^{1/p}$. Again,

$$\mathrm{KL}(P_1^{\otimes n} \| P_0^{\otimes n}) = n \, \mathrm{KL}(P_1 \| P_0) \leq n \, \chi^2(P_1 \| P_0) = 4n\delta^2.$$

By Pinsker's inequality (Tsybakov, 2009, Lemma 2.5), note that,

$$\mathrm{TV}(P_1^{\otimes n}, P_0^{\otimes n}) \leq \sqrt{\frac{1}{2} \mathrm{KL}(P_1^{\otimes n} \| P_0^{\otimes n})} = 2\delta\sqrt{n} = 1/4,$$

if $\delta = \frac{1}{8\sqrt{n}}$. Thus from equation 15.14 of Wainwright (2019), observe that,

$$\inf_{\hat{\mu}} \sup_{\mu \in \Pi_{\mathbb{M}}} \mathbb{E}_\mu \|\hat{\mu} - \mu\|_{\mathbb{H}^\beta} \gtrsim \delta \asymp 1/\sqrt{n}. \tag{8}$$

$$\text{Similarly,} \quad \inf_{\hat{\mu}} \sup_{\mu \in \Pi_{\mathbb{M}}} \mathbb{E}_\mu \mathbb{W}_p(\hat{\mu}, \mu) \gtrsim \delta^{1/p} \asymp n^{-\frac{1}{2p}}. \tag{9}$$

The result now follows from combining (6) and (8). The minimax rate for the Wasserstein distance follows from combining (7) and (9).

## 4 Conclusion

This paper aimed to study the fundamental question of whether GANs are optimal in providing accurate estimates of target distributions, especially when the data exhibit a low-dimensional structure. This notion of low dimensionality is characterised by the so-called Minkowski dimension. In scenarios where $n$ independent and identically distributed samples are available from a target distribution on a set $\mathbb{M}$, the convergence rate for any estimator is bounded by $\Omega\left(n^{-\frac{\beta}{d_{\mathbb{M}}-\delta}} \vee n^{-1/2}\right)$ in the $\beta$-Hölder IPM and $\Omega\left(n^{-\frac{1}{d_{\mathbb{M}}-\delta}} \vee n^{-\frac{1}{2p}}\right)$ in the Wasserstein $p$-metric. When the support is regular in the Minkowski sense, the convergence rates for GANs closely resemble this lower bound (in the $\beta$-Hölder IPM), when the networks are properly chosen. Some future results in this direction might render fruitful avenues for understanding similar lower bounds, especially with a different notion of the dimensionality of the data distribution, such as the Wasserstein dimension (Weed & Bach, 2019). Further, while GANs are known to achieve optimal rates in the Wasserstein-1 distance, it remains an open question whether this result extends to Wasserstein-$p$ distances for $p \geq 2$.

## Acknowledgement

I would sincerely like to thank Prof. Peter L. Bartlett for his invaluable insights, thoughtful discussions and constant guidance. The author also gratefully acknowledges the support of the NSF and the Simons Foundation for the Collaboration on the Theoretical Foundations of Deep Learning through awards DMS-2031883 and #814639, the NSF's support of FODSI through grant DMS-2023505, and the support of the ONR through MURI award N000142112431.

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
