# OpenReview forum: "Minimax Lower Bounds for Estimating Distributions on Low-dimensional Spaces"
_TMLR — Accepted by TMLR_

### Review · Reviewer_ZUsU · 2024-11-29

**Summary Of Contributions:**

This paper studies the problem of how well one can estimate unknown distributions with low-dimensional structures. The authors prove lower bounds for the minimax rates in H\"older IPM and p-Wasserstein metric, and show that the rates depend on the Minkowski dimension of the support of the distribution rather that the ambient dimension. Combining with recent results on the convergence rates of GANs, it is shown that GANs are nearly minimax optimal in estimating low-dimensional distributions.

**Audience:**

Yes

**Claims And Evidence:**

Yes

**Requested Changes:**

1. The convergence rate of GANs presented in this paper is slightly different from the reference [Huang et al. JMLR 2022] (A $\delta$-term is added to the rate). The statement is correct, but the authors should explain the relation and difference of these two bounds (I think the difference is from the different assumptions on the covering number). This is also related to the $\delta$-term in the lower bound. It is said in the third paragraph of Page 5 that the $\delta$-term is an artifact of the definition of the Minkowski dimension. I think it would be better to write this paragraph as a remark and discuss how assumptions on the covering number affect the upper and lower bounds for the convergence rate.

2. In the proof of Theorem 7, the authors apply Theorem 15.2 and equation 15.14 of [Wainwright 2019] to derive lower bounds (6)-(9). Can the authors explain how the results of [Wainwright 2019] are used here. What are the function $\Phi$ and metric $\rho$? Please also check whether ``Theorem 15.2'' is the correct reference.

3. Minor issues and typos:

(1) Page 5, Paragraph 4, ``t is well known that We recall that we call a set''.

(2) Page 6, the last equality in the Proof of Lemma 8, I think $\delta = sep(\Xi)/3$ rather than $\delta = sep(\Xi)$.

(3) Page 6, the second paragraph of the Proof of Theorem 7, I think $\omega \in$ \{0,1\}$^k$ should be $\omega \in$ \{0,1\}$^{\lfloor k/2 \rfloor}$.

**Strengths And Weaknesses:**

Strengths: The paper is well-written. The results are solid.

Weaknesses: I think some of the statements and proofs need to be explained in more detail for the readers' convenience. See [Requested Changes] for examples.

---

> ### Author Response · Authors · 2024-12-12
> **Response to Reviewer ZUsU**
>
> # Reviewer ZUsU
> We thank the reviewer for their overall positive feedback on our work and thoughtful and constructive suggestions. We respond to the points raised by the reviewer as follows:
>
> **Comparison with the Rates from Huang et al. (2022)**: Indeed, as the reviewer rightly pointed out, the slight difference in the reported rates in Huang et al. (2022), compared to the one we mention, stems from the fact that Huang et al. (2022) assume that $\mathcal{N}(\epsilon, \mathbb{M}, \ell_\infty) \precsim \epsilon^{-d}$, for some $d$, while we use the definition of the upper Minkowski dimension of $\mathbb{M}$, which uses a $\lim \sup$, ensuring that for any positive $\delta$,  $\mathcal{N}(\epsilon, \mathbb{M}, \ell_\infty) \precsim \epsilon^{-(d+\delta)}$. Following the reviewer's suggestion, we have mentioned this as a remark in Remark 1 on page 6 of the revised manuscript.
>
> "**Remark 1** ($\delta$-term in Theorem 7).  We observe that the $\delta$-term is an artifact of the definition of the Minkowski dimension. If $\mathcal{N}(\epsilon, \mathbb{M}, \ell_\infty) \succsim \epsilon^{-\underline{d}}$, for some $\underline{d} \in (0, D]$, i.e. when the limit in the computation of the lower Minkowski dimension can be achieved exactly, following the proof of Theorem 7 (see Section 3), we note that under the same assumptions $\mathfrak{M}_n  \succsim n^{-\beta/\underline{d}} \vee n^{-1/2}$, for $n$ large.
>
>  The $\delta$-term in the lower bound is only an artifact of the definition of the lower Minkowski dimension and can be removed by assuming the lower bound for the covering number. Further, if one assumes that $\mathcal{N}(\epsilon, \mathbb{M}, \ell_\infty) \precsim \epsilon^{-\bar{d}}$, then the analyses by Huang et al. (2022) shows that the error rate for GANs for estimating a distribution $\mu$, supported on $\mathbb{M}$ scales as $(n^{-\beta/\bar{d}} \vee n^{-1/2}) \log n$ (see Theorem 19 of Huang et al. (2022)). Thus, when $\mathcal{N}(\epsilon, \mathbb{M}, \ell_\infty) \asymp \epsilon^{-d}$, the error rates for GANs match the minimax rate except for an excess $\log$-factor in the number of samples."
>
> **Clarification on Choices of $\Phi $ and $\rho$**: We thank the reviewer for raising the issue. Indeed the correct reference should be Theorem 15.12 instead of 15.2. We take the choices of $\Phi(x)=x$ and $\rho(P,Q) = \|P-Q\|_{\mathbb{H}^\beta}$. We have now mentioned this on page 8 of the manuscript.
>
> **Minor Typos**: We thank the reviewer for meticulously going through the manuscript. We have now weeded out the typos you mentioned along with others for ease of readability.

---

> > ### Comment · Reviewer_ZUsU · 2024-12-15
> > **Response to the authors**
> >
> > Thank you for your response and clarification. I have two comments on the revised paper. In page 2, the last paragraph above Section 1, the parameter $W$ in [Huang et al. 2022] is the width of the network rather than the number of weights (so $W^2L$ is roughly the number of weights). In the last sentence of the proof of Lemma 8, one should take $\pi_1 = 2a/ 3^\beta$.

---

> > > ### Author Response · Authors · 2024-12-18
> > >
> > > We sincerely thank the reviewer for bringing these points to our attention. We have now corrected them in the revised manuscript.

---

### Review · Reviewer_RHKH · 2024-12-04

**Summary Of Contributions:**

In the paper, lower bounds for the minimax estimation rate of probability distributions in terms of both the $\beta$-Hölder metric and the p-Wasserstein distance are derived. Special attention is given to data with low-dimensional support and it is shown, by comparing to results from previous work, that in such cases GANs achieve almost optimal convergence rates.

**Audience:**

Yes

**Claims And Evidence:**

Yes

**Requested Changes:**

See weaknesses above.


Minor issues:

Section 2, first paragraph "... best performing estimator that achieves the maximum risk": maximum -> minimum

Theorem 7: "... exists and" -> "... exists an"

Page 5: "the sample mean almost achieves this minimax optimal rate": sample mean -> empirical distribution

Page 5: "it is well known that We recall that"

Page 5: d-Hausdroff -> d-Hausdorff

Page 6, last equation: ")" missing before ^2 at the end

**Strengths And Weaknesses:**

Strengths:

*) Deriving theoretical guarantees for machine learning algorithms is important.

*) The paper is well-structured and well-written. While I was not able to check the full proof in detail, it appeared reasonable when looking through it.

Weaknesses:


*) Throughout the paper, I noticed quite a few typos (see the requested changes below). Hence, the paper should be carefully proofread again.

*) While I understand that this is a theory paper, it would have been interesting to see a brief discussion on bounds that have been observed empirically for the chosen metrics. If such results exist, it would be good to briefly discuss this aspect, for example in the related work section.

*) On page 2, at the end of the introduction it says that "... GANs almost match this rate, when the networks are properly chosen". What does it mean for the networks to be properly chosen? This point should be made more precise.

---

> ### Author Response · Authors · 2024-12-12
> **Response to Reviewer RHKH**
>
> # Reviewer RHKH
> We thank you for the overall positive reception of our work and constructive feedback. Following your suggestions, we have made improvements to our manuscript, details of which are as follows.
>
> **Typos**: We thank the reviewer for taking the time to go through the paper in such details. We have now implemented your suggestions and have also thoroughly proofread the manuscript to weed out the typos.
>
> **Empirical Results from the Literature**: While our work primarily focuses on theoretical analysis, we agree that a brief discussion of empirical results would provide additional context to the reader. There have been some empirical analyses, supported by theory that show that the error rates (in terms of the number of training samples) for different deep learning problems depend on the intrinsic data dimension. We have added the following paragraph in the related works section on page 3.
>
> "In addition to the mathematical results (Huang et al., 2022) that the error rates for GANs depend on the intrinsic data dimension, empirical results also indicate such phenomenon occurs in practice. For example, the recent works by Chakraborty & Bartlett (2024a) show that the test error rates in the Wasserstein-1 distance for Wasserstein Autoencoders (Tolstikhin et al., 2018) depend only on the Minkowski dimension of the data support. Similar results have also been established both theoretically and empirically for deep
> regression (Nakada & Imaizumi, 2020) and federated learning models (Chakraborty & Bartlett, 2024b)."
>
> *References*
>
> Saptarshi Chakraborty and Peter Bartlett. A statistical analysis of Wasserstein autoencoders for intrinsically
> low-dimensional data. In The Twelfth International Conference on Learning Representations, 2024a. URL https://openreview.net/forum?id=WjRPZsfeBO.
>
> Saptarshi Chakraborty and Peter L. Bartlett. A statistical analysis of deep federated learning for intrinsically low-dimensional data, 2024b. URL https://arxiv.org/abs/2410.20659.
>
> Ryumei Nakada and Masaaki Imaizumi. Adaptive approximation and generalization of deep neural network with intrinsic dimensionality. Journal of Machine Learning Research, 21(174):1–38, 2020. URL http://jmlr.org/papers/v21/20-002.html.
>
> **Network Choices**: We agree with the reviewer that clarification of the network choices would further enhance the readability of the paper. To that end, we have added the following paragraph by the end of the introduction on page 2, following the reviewer's suggestion.
>
> "{In particular, Huang et al. (2022) showed that when the generator and discriminators are realized by a feed-forward neural network with ReLU activation, respective depth $L_\text{gen}$, $L_\text{dis}$ and the number of weights $W_\text{gen}$, $W_\text{dis}$, then one can choose $W_\text{gen}^2 L_\text{gen} \precsim n$ and $W_\text{dis} L_\text{dis} \precsim n^{D/(2\bar{d}_\mathbb{M} +2 \delta)} \log^2 n$
>
> ($D$ denotes the dimension of the data space) to ensure that the error rate for the GAN estimate of the distribution scales as $\mathcal{O}(n^{-\beta/(\bar{d}_\mathbb{M} +\delta )} \vee n^{-1/2} \log n)$. In this case, the discriminator network has to satisfy a regularity condition in terms of its maximum Lipschitz constant. We refer the reader to Theorem 19 of Huang et al. (2022) for more details."

---

> > ### Comment · Reviewer_RHKH · 2024-12-20
> >
> > Thank you for the response, my concerns have been addressed.

---

### Review · Reviewer_Ptbn · 2024-12-07

**Summary Of Contributions:**

The paper gives the minimax lower bound  of GNAS in terms of low dimension structure for the measure.

**Audience:**

Yes

**Claims And Evidence:**

Yes

**Requested Changes:**

1. Could the authors add a table which compares lower bound and upper bound of GAN under different settings? In that way, the readers can have a better understanding of the literature and the contributions in this paper.
2. Could the authors clarify, if GAN is still optimal, in terms of the Wasserstein distance? That is to say, what is the upper bound of GAN in terms of Minkowski dimension and Wasserstein distance?

**Strengths And Weaknesses:**

Strength: The paper gives the minimax lower bound of GAN in terms of low dimension structure. The paper is well-written and is easy to follow

Weakness:

1. The contribution seems to be incremental, especially compared with Tang, Rong, and Yun Yang. "Minimax rate of distribution estimation on unknown submanifolds under adversarial losses." The Annals of Statistics 51.3 (2023): 1282-1308. It will be better if the author could point out the technical challenges to increase the contribution.

2. The comparison with the literature is weak. Some tables which could contain all the upper/lower bounds of GAN could make the comparison more concrete.

---

> ### Author Response · Authors · 2024-12-12
> **Response to Reviewer Ptbn**
>
> # Reviewer Ptbn
> We thank the reviewer for their thoughtful comments and suggestions. We respond to the individual points raised as follows:
>
> **Comparison with Tang and Yang (2023)**:  Despite the overlap in the main theme of the problem with Tan and Yang (2023), there are significant differences in problem settings and assumptions.  In contrast to the work of Tang and Yang (2023), our analysis does not impose a smooth manifold structure on the data support, allowing for highly non-smooth and irregular geometries. Moreover, we do not assume the existence of a density for the distribution on this potentially irregular low-dimensional support, enabling us to accommodate singular distributions on this structure within our framework. While both our work and the one by Tang and Yang (2023) leverage the principles of minimax error bounds via Fano's method, their reliance on smooth manifolds and the existence of smooth densities simplifies the corresponding testing problem for rate derivation. In contrast, our approach circumvents these assumptions by constructing point measures on the low-dimensional structure, enabling a more general analysis under minimal assumptions. Further, while Tang and Yang(2023) derive the minimax error bounds for $\beta$-H\"{o}lder IPMs, our analyses also cover the Wasserstein $p$-distances. Following the reviewer's suggestion, we have included the following discussion in the related works section of the paper in Section 1.1 to emphasize the novelty of our contributions.
>
> "In contrast to the work of Tang and Yang (2023), our analysis does not impose a smooth manifold structure on the data support, allowing for highly non-smooth and irregular geometries. Moreover, we do not assume the existence of a density for the distribution on this potentially irregular low-dimensional support, enabling us to accommodate singular distributions on this structure within our framework. While both our work and the one by Tang and Yang (2023) leverage the principles of minimax error bounds via Fano's method, their reliance on smooth manifolds and the existence of smooth densities simplifies the corresponding testing problem for rate derivation. In contrast, our approach circumvents these assumptions by constructing point measures on the low-dimensional structure, enabling a more general analysis under minimal assumptions. Further, while Tang and Yang(2023) derive the minimax error bounds for $\beta$-H\"{o}lder IPMs, our analyses also cover the Wasserstein $p$-distances."
>
> **Table of comparing recent literature**: We agree with the reviewer that adding a table of comparison between different upper and lower bounds for distribution estimation will further enhance the readability of the paper and highlight the contributions of this work against the state-of-the-art. We have included a comparative analysis in Table 1 of the revised manuscript, following the reviewer's suggestion.
>
> **Optimality of GANs in Wasserstein-$p$ distance**: We thank the reviewer for the question. While, GANs are known to achieve minimax optimal rates in the Wasserstein-1 distance, to the best of our knowledge, it remains an open question whether this result extends to Wasserstein-$p$ distances for  $p \ge 2$. Our intuition is that the problem becomes more challenging for higher-order Wasserstein distances, as they are not naturally expressed as integral probability metrics, which are central to the GAN framework. We agree with the reviewer that this is an interesting direction for future research. In response, we have highlighted this open problem in the conclusion section of the paper (Section 4).

---

### Author Response · Authors · 2024-12-12
**Response to Reviewers**

We thank the Action Editor for the opportunity to make revisions, and the anonymous reviewers for their insightful comments and critiques. We have provided a thoroughly revised draft, addressing the reviewers' comments. Major changes are marked in blue throughout the manuscript. We hope you will find our response thorough and complete, leading to a improved manuscript, and thank you for the potential opportunity to share our findings with the TMLR readership.

---

### Decision · Action_Editor_2Sp2 · 2025-01-12

**Recommendation:** Accept as is

**Comment:**

The paper gives solid theoretical analysis to derive minimax lower bounds. Interestingly, the lower bounds depend on the low-dimensional structure of the data which is useful for high-dimensional problems. The paper presents clear tables and discussions to compare the derived results with existing results, e.g., removing some assumptions and more general results. All the reviewers are satisfied with the revision and the discussions.

Minor comment:
- page 3: "Tang and Yang(2023)" should be "Tang and Yang (2023)"

**Audience:**

Deriving minimax lower bounds is useful to understand the limitations of existing methods for density estimation. The derived results should be interesting to the TMLR community.

**Claims And Evidence:**

This paper develops lower bounds for density estimation by GANs. The lower bounds depend on the lower Minkowski dimension of the support of the data distribution. This matches the existing upper bounds if the lower Minkowski dimension is equal to the upper Minkowski dimension. The paper gives solid theoretical analysis to derive minimax lower bounds by Fano's method.